# Examination of marketing mix performance in relation to sustainable development of the Poland's confectionery industry

**Pawel Tadeusz Kazibudzki**[1]☯*, **Tomasz Witold Trojanowski**[2]☯

**1** Faculty of Economics and Management, Opole University of Technology, Opole, Republic of Poland,
**2** Department of Management, Jan Kochanowski University in Kielce, Kielce, Republic of Poland

☯ These authors contributed equally to this work.
* p.kazibudzki@po.edu.pl, poczta@gmail.com

Examination of marketing mix performance in
relation to sustainable development of the Poland's
confectionery industry. PLoS ONE 15(10):
e0240893. https://doi.org/10.1371/journal.
pone.0240893

SPAIN

**Data Availability Statement:** All relevant data are
within the manuscript and its Supporting
Information files.

## Abstract

The conventional concept of marketing mix does not take into account the idea of sustainable development. The basic objective of this examination is to analyze and evaluate the performance of selected marketing mix elements from the perspective of the Poland's confectionery industry's sustainable development. The questionnaire survey was designed for this purpose. The purpose of the research questions was to evaluate a degree of development for selected elements of marketing mix from the perspective of sustainable development of the Poland's confectionery industry. Thus, a novel development ratio based on the distance from exemplary performance was proposed. Next, a seminal approach to pairwise comparisons technique was applied for the importance evaluation of each survey question in order to provide a weighted average Mean Development Ratio ($MdeR$) for each element of marketing mix. In this process the seminal methodology for pairwise comparisons was applied i.e. a non-heuristic approach to pairwise comparisons technique with verifiable accuracy and reliability. In consequence, assuming that all elements of marketing mix have some designated importance in the process of sustainable development, a total weighted average $MdeR$ for performance of all elements of marketing mix was computed and evaluated. Noticeably, the total weighted average $MdeR$ for performance of all elements of marketing mix cannot be considered as satisfactory from the perspective of sustainable development of the Poland's confectionery industry.

## Introduction

Excessive exploitation of natural resources, consumption of fuels, energy and water, increased waste generation and harmful substances, together with the progressing growth of the world's population—contribute to irreversible degradation of the natural environment and deterioration of societies' life quality [1]. Environmental degradation has a negative impact on the health condition of modern societies, and also reduces the chances of future generations' development. Food industry enterprises also take part in this interference as they are perceived

**Funding:** The article processing charge was funded by Opole University of Technology and Jan Kochanowski University in Kielce. The funders had no role in study design, data collection and analysis, decision to publish, or preparation of the manuscript. No additional external funding was received for this study.

**Competing interests:** The authors have declared that no competing interests exist.

as not very innovative [2], unlike enterprises operating in other sectors such as the automotive or IT industry. To meet existing environmental and social problems, more attention should be focused on the concept of sustainable development [3]. By promoting pro-ecological and pro-social lifestyle, including consumer behavior, various types of environmental and social organizations aim to educate contemporary societies in this area [4,5]. The concept of sustainable development influences the business philosophy of business entities behavior, shaping their actions in the area of management and marketing.

Responsible marketing activities have an important role in reducing the emerging problems caused by the business operations of food industry enterprises. Marketing is primarily associated with sales and activities encouraging customers to buy various types of goods and services. For this reason, marketing is at odds with the concept of sustainable development [6,7]. Marketing mix was also criticized. The conventional concept of marketing mix does not take into account the idea of sustainable development, apart from social and environmental aspects. The classic marketing mix has been criticized e.g.Goi [8] or Glavas and Mish [9]. Möller [10] and Popovic [11] also joined the group of critics. A marketing composition consisting of four elements represents the interests of the producer rather than the buyer i.e. the consumer of the product. The opposite of conventional marketing, including marketing mix [12,13] is sustainable marketing [14,15] which is a sub-area of sustainable management. The essence of sustainable marketing is not only achieving food companies' economic goals, but also environmental and social goals [16]. An important role in responsible marketing policy of food industry enterprises is therefore the concept of sustainable marketing mix reflecting product, price, distribution (place), and promotion mix. The use of marketing mix instruments supporting sustainable development by food industry enterprises is particularly important because production processes significantly affect the state of the natural environment, thus contributing to the emergence of environmental and social problems. Sustainable marketing mix assumes the introduction of food products onto the market that will properly meet the needs and desires of buyers with minimal impact on the natural environment.

The purpose of the article is to examine and determine the forms and ways of implementing marketing instruments into the field of sustainable marketing mix for Poland's confectionery industry. In addition to the purpose of the study, the research hypothesis was also evaluated. The hypothesis was worded in the following way: Marketing activities in the field of individual marketing mix instruments, undertaken by enterprises operating in the confectionery industry are simply conventional, which in consequence entails that these enterprises do not implement a marketing mix based on the principles of sustainable development.

The data was obtained from the authors' own primary surveys, for the years 2017–2018, and its elaboration was supported by *Microsoft Excel Software*. The data was also examined within *QSP Multi Criteria Decision Support Tool* [17] aiding a seminal methodology for pairwise comparisons of Tomashevskii&Tomashevskii [18]. The article is organized as follows: After the introduction (Section 1), research methodology and research effects are discussed (Section 2). First, the survey's concept and the examination method are outlined and their design is explained (Subsection 1), then survey's results are presented and their fundamental input to the examination process is designated (Subsection 2). Then the examination's seminal method is described and its outcome delivered, analyzed and evaluated (Subsection 3). Discussion (Section 3) follows Section 2 and is devoted to the review of pertinent literature, and the partial and total examination results of the research paper. The article is closed by conclusions (Section 4) which summarize the examination efforts with final remarks.

## Research methodology and effects

### Methodology outline

Following the basic objective of this research i.e. to analyze and evaluate the performance of marketing mix elements from the perspective of sustainable development of the confectionery industry, an anonymous questionnaire survey was designed and submitted to 74 randomly selected companies operating within the Poland's confectionery industry. Thirty-three questions were asked in the questionnaire which anonymously examined performance of four basic marketing mix elements (product, price, place, and promotion mix) from the perspective of sustainable development. The survey questionnaire—S1 Table, contained closed questions concerning the particular company's approach towards sustainable marketing mix issues. Questions arose as a result of a brainstorming conducted by 9 experts. Work on the final form of the questions included in the research questionnaire was preceded by a pilot study phase when the face and content validity of the questionnaire were examined. The pilot study was a kind of sample before the actual research, which is particularly recommended to be carried out due to the selection of research techniques and tools [19,20]. The pilot study aimed to verify the meaning and clarity of the questions and confirm the agreement of the researcher's intentions with respect to the answers given as well to verify the reliability (stability) and homogeneity (internal consistency) of the questionnaire [11,20,21]. As a result of the research, some questions and answer options were clarified. It has also been suggested to provide a more concise statement on how to respond. After the modification, the questionnaire took its final form. The questionnaire occurred to be stable with correlation coefficient r $\geq$ 0.70. Also, typically used during scale development with items that have several response options coefficient alpha was higher than 0.70 what indicates good reliability of the questionnaire [19,20,22].

The objective of the research questions was to examine the degree of development for all elements of a marketing mix in relation to the confectionery industry sustainable development. For that purpose it was decided to apply the *Development Ratio(deR)* based on the distance from exemplary performance i.e. the gradient method [23–26]. The ratio detailed construction and its description is presented in the section "Survey's Concept and Results". This ratio constitutes the measure of marketing mix elements performance which was diagnosed with the application of the questioning technique. Each question was evaluated on the basis of Likert's seven degrees scale [27] which is a common technique for the measurement of attitudes in social research [22,28,29]. Of course it can be discussed why seven degrees scale was applied, why not five or eleven degrees scale. There are different views on this matter [30]. However, taking into consideration the objective of this research and the limited channel capacity of humans [31–33], Likert's seven degrees scale was considered here as the optimal. On the basis of respondents evaluations, the development ratios for all questioned areas of marketing mix elements were calculated. Then, their importance was weighted. In order to precisely establish the weights values, the pairwise comparison method was used in its novel form [17,18].

Basically, two kinds of measurement techniques can be distinguished for this purpose i.e. the relative measurement technique and the absolute measurement one. In the case of a large number of alternatives, the latter technique is often utilized [33–35], which is often also called the rating approach. This approach consists of defining *intensities* of achievement or preference for criterion or criteria in a model. These *intensities* are used in place of alternatives in the process of their evaluation in the first stage. For example, instead of using pairwise comparisons of relative preference for specific alternatives with respect to some criterion [36], one can compare the relative preference of a nonspecific alternative that completely fulfills that criterion to some other alternative that fulfills that criterion only partially. Such pairwise

comparisons result in measures of preference for the *intensities* which possess the ratio-scale property. Then, in the second stage, each alternative (in this research, every single development ratio) is evaluated through its intensity in relation to each criterion (in this research there is only one, the same criterion for each element of sustainable marketing mix which is 'the performance').

In the survey, there are totally 33 alternatives (development ratios established on the basis of survey's questions answers) diagnosing the performance of four elements of the sustainable marketing mix i.e. sustainable: product, price, place, and promotion mix. Thus, weights for 12 development ratios needed to be established for the performance evaluation of the sustainable product, weights for 6 development ratios needed to be established for the performance evaluation of the sustainable price, weights for 10 development ratios needed to be established for the performance evaluation of the sustainable distribution, and weights for 5 development ratios needed to be established for the performance evaluation of the sustainable promotion mix. For this purpose the rating approach was applied. Five intensities for pairwise comparisons were proposed i.e. extreme, high, moderate, slight, and tiny. Again, it can be disputed why as many as five intensities were proposed. However, taking into consideration the objective of the pairwise evaluation, the possible errors in this process, and the limited channel capacity of humans [31–33], this particular number of intensities was considered here as the optimal. All the necessary computations were processed in the *QSP Multi Criteria Decision Support Tool* [17] aiding a seminal methodology for pairwise comparisons [18]. The objective of the approach was firstly, to evaluate the importance of each survey's question in order to provide a weighted average Mean Development Ratio *(MdeR)* for each element of a sustainable marketing mix i.e. sustainable product–$waMdeR^{(pt)}$, sustainable price–$waMdeR^{(pc)}$, sustainable distribution (place)–$waMdeR^{(pl)}$, and sustainable promotion mix–$waMdeR^{(pr)}$; and secondly, to compute and evaluate a total weighted average *MdeR* for performance of all elements of a marketing mix i.e. *waMdeR(t)*. The full description of the process and its results is presented in the section "Examination's Method and Outcome".

## Survey's concept and results

The confectionery industry, apart from the meat, vegetable and fruit industry, clearly affects the condition of natural environment [37–39]. The main ecological threats arising in the production of confectionery include the significant use of water in production processes and the associated waste water emissions, generating product waste with particular emphasis on organic waste, emission of harmful substances into the atmosphere, including dusts, gases and odors, energy consumption, as well as noise emitted by machines and devices. Reference should also be made to the impact of confectionery consumption on human health. The vast majority of confectionery products contain sugar, which is one of the main causes of diabetes. Excessive and uncontrolled consumption of sugar found in sweets leads to other diseases, i.e. overweight, obesity and cardiovascular diseases. Other serious hazards arising from high sugar consumption include the formation of some types of cancer, e.g. pancreas. Eating sweets is closely related to dental health. Dental caries and premature tooth loss are more often observed among children eating sweets several times a day than those who rarely consume these products. In addition to sugar content, confectionery products also contain large amounts of fat as well as saturated fatty acids and acids in a trans configuration. The consumption of fat and trans fatty acids conduce the development of heart disease, cardiovascular disease, atherosclerosis, diabetes, obesity, cancer, as well as impaired function of the immune system [40–46].

Thus, the subjects of the research are enterprises recognized by the Polish Classification of Business Activity as PKD 10.72.Z (the production of rusks and biscuits, the production of

conserved pastry articles and cakes) and PKD 10.82.Z (the production of cacao, chocolate and confectionery). It was found that the Polish Central Statistical Bureau's database REGON lists a few hundred of such businesses. However, the REGON database is not fully trustworthy as it has a declarative nature i.e. first of all, many companies registered in the REGON database could already have liquidated their business activity, or could have temporarily suspended their business activity; and secondly, they could have declared a few areas of their business activity (just in case) what does not entail they operate in all of them. It is why the authors decided to examine the issue from different sources of information. Two other more credible databases were taken into consideration i.e. the Central Record and Information about Business Activity (CEIDG database) and the Domestic Judiciary Register (KRS database). Furthermore, some additional unpublished data of the Polish Central Statistical Bureau concerning the Polish confectionery industry was taken into consideration as well. It was established that there are 223 enterprises in Poland which constitute the core of Poland's confectionery industry.

It should be emphasized that the confectionery industry in Poland is diversified i.e. companies operating in it differ from the perspective of their production profile, their market position and market potential. Among firms operating in the industry, one can distinguish companies which should be described as global organizations e.g. Mondelēz International Inc., Nestle, Mars, and Ferrero Corporations, as well small and medium sized domestic family businesses. Noticeable disproportion and spread among analyzed organizations can be justified by different goals set up by particular companies. Large foreign and domestic corporations that offer a dozen or tens of diversified products, focus mainly on sales increase and securing the best market share and market position. On the other hand, smaller businesses focus on the composition of healthy, exceptional and distinguished products.

Due to limited financial resources for the project, it was decided to examine only a part of the population. In the case of dependent sampling, and designated level of significance and accuracy of the results, the minimal size for the sample is calculated on the basis of the following Eq (1).

$$n \geq \frac{N\mu_{\alpha/2}^2}{4(N-1)d^2 + \mu_{\alpha/2}^2} \tag{1}$$

Thus, for assumed level of significance $\alpha = 0.1$ that designates $\mu_{\alpha/2} = 1.645$, and assumed level of accuracy $d \leq 0.08$ which guaranties ±8% of error margin that can be considered as the relatively decent level of credibility for an estimate, and taking into account the size of target population i.e. $N = 223$, we have $n \geq 72$. In the research, the randomly drawn sample of $n = 74$ companies operating in the confectionery industry in Poland was examined. Its structure present Tables 1–5.

Table 1. Legal forms of examined companies.

| Legal form of the business | Number of firms | Percent of firms |
|---|---|---|
| Public limited company | 2 | 2.7% |
| Limited liability company | 33 | 44.5% |
| Limited partnership | 1 | 1.4% |
| General partnership | 25 | 33.8% |
| Sole proprietorship | 13 | 17.6% |

**Table 2. Number of employees.**

| 0–49 | | 50–249 | | 250 and more | |
|---|---|---|---|---|---|
| No. of firms | Percent | No. of firms | Percent | No. of firms | Percent |
| 12 | 16.2% | 42 | 56.8% | 20 | 27% |

**Table 3. Position of the respondent.**

| Managerial position | | Non-managerial position | |
|---|---|---|---|
| No. of respondents | Percent of respondents | No. of respondents | Percent of respondents |
| 50 | 67.6% | 24 | 32.4% |

**Table 4. Respondents' professional experience (period of service in years).**

| 5 or less | | 6–10 | | 11–20 | | 21–30 | |
|---|---|---|---|---|---|---|---|
| Number | Percent | Number | Percent | Number | Percent | Number | Percent |
| 27 | 36.5% | 22 | 29.7% | 22 | 29.7% | 3 | 4.1% |

Following the basic objective of this research i.e. to analyze and evaluate the performance of marketing mix elements from the perspective of sustainable development of the Poland's confectionery industry, a questionnaire survey was designed and submitted to 74 randomly selected companies operating within confectionery industry in Poland. Thirty-three questions were asked in the questionnaire which examined performance of four basic marketing mix elements (product, price, place, and promotion mix) from the perspective of sustainable development. The exemplary survey's questionnaire is attached to this research paper as S1 Table. The set of data gathered during the survey is also enclosed to this research—S2–S4 Tables.

The objective of the research questions was to examine the degree of development for all elements of a marketing mix from the perspective of the Poland's confectionery industry sustainable development. For that purpose it was decided to apply a development ratio based on

**Table 5. Provinces where the businesses operate.**

| Province | Number of firms | Percent of firms |
|---|---|---|
| Lower Silesia | 7 | 9.5% |
| Podlasie | 5 | 6.8% |
| Pomeranian | 1 | 1.4% |
| Silesia | 11 | 14.8% |
| Greater Poland | 9 | 12.1% |
| Kuyavian-Pomeranian | 5 | 6.8% |
| Lublin | 4 | 5.3% |
| Lubusz | 2 | 2.7% |
| Lodz | 6 | 8.1% |
| Lesser Poland | 7 | 9.5% |
| Mazovian | 11 | 14.8% |
| Opole | 3 | 4.1% |
| Subcarpathian | 3 | 4.1% |

the distance from exemplary performance i.e. the gradient method [23–26]. The ratio construction can be described as follows.

Assumed that the evaluated object $X_i$ is denoted by the vector $X_i = (x_{i1}, x_{i2}, x_{i3}, \ldots, x_{in})$ where $x, i, n \in N$. The exemplar i.e. the object described by variables which are considered as desired, is denoted by the vector $\hat{X}_i = (\hat{x}_{i1}, \hat{x}_{i2}, \hat{x}_{i3}, \ldots, \hat{x}_{in})$ where $\hat{x}, i, n \in N$, and the anti-exemplar, i.e. the object described by variables which are considered as undesired, is denoted by the vector $\tilde{X}_i = (\tilde{x}_{i1}, \tilde{x}_{i2}, \tilde{x}_{i3}, \ldots, \tilde{x}_{in})$ where $\tilde{x}, i, n \in N$. Then, the development ratio ($deR_{ik}$) can be defined in a form of the following Eq 2.

$$deR_{ik} = \frac{x_{ik} - \tilde{x}_{ik}}{\hat{x}_{ik} - \tilde{x}_{ik}} \tag{2}$$

where $k \in \{1, 2, 3, \ldots, n\}$, the numerator designates Euclidian distance of the examined object's element from its anti-exemplary value, and the denominator designates Euclidian distance of the element's exemplary value from its anti-exemplary value. In this research $x_{ik} \in \{0, 1, 2, 3, 4, 5, 6\}$, $\hat{x}_{ik} = 6$, and $\tilde{x}_{ik} = 0$.

Noticeably, $deR_{ik} \in [0,1]$, and the closer it is to unity, the higher is the development degree of the evaluated element in the particular object. In this research, mean development ratios were applied Eq (3) which exemplary calculations are explained in details when survey results are presented later on in this section.

$$MdeR_i = \frac{1}{n}\sum_{k=1}^{n} deR_{ik} \tag{3}$$

where $n$ denotes the total number of answers collected in the survey i.e. $n = 74$.

Basically, performance of four fundamental elements of marketing mix was examined: product, price, place (distribution), and promotion mix from the perspective of their performance in relation to sustainable development. Each element of the marketing mix was evaluated from the perspective of a few areas. Each area was represented by a particular question asked of respondents. The survey comprised in total $i = 33$ questions which examined entire marketing mix performance from the perspective of the sustainable development concept. All questions were elaborated by a group of nine experts who also evaluated each question from the perspective of its total contribution to the problem (details of this evaluation will be presented later in this research in its section entitled *Examination's Method and Outcome*). The experts were appointedon the basis of their experience from nine deliberately selected successful businesses operating in the territory of Poland (Table 6 presents the experts characteristics).

Answers in the questionnaire (S1 Table) were designed and coded in the seven degrees version of Likert's scale [27] which guaranties quite decent accuracy of judgments. Tables 7–10 present the result of the survey together with examined areas ($Q_i$) evaluations reflected by their mean development ratios $MdeR_i$. Answers' characteristics in Tables 7–10 are presented in alternant squares, shaded and not shaded. There are four numbers in each square. The upper left corner in each square informs of the number of particular answers for the particular question that relates to the examined area ($Q_i$). The upper right corner in each square provides a percentage share of particular answers for the particular question in the total number of answers i.e. $n = 74$. The lower left corner in each square provides a code for the particular answer in this research $x_{ik} \in \{0, 1, 2, 3, 4, 5, 6\}$. The lower right corner in each square presents a product of the answer's code and a number of particular answers for the particular question. In this research then, the quotient of the products' sum for the particular question and the maximal value for that sum i.e. $n \cdot \hat{x}_{ik} = 74 \cdot 6 = 444$, provides the $MdeR_i$ value.

**Table 6. Experts' characteristics.**

| No | Legal form of the business | Firm's employment | Province of firm's activity | Expert's business experience | Expert's status |
|---|---|---|---|---|---|
| 1 | Limited partnership | 50–249 | Silesian | 16 years | Entrepreneur-manager |
| 2 | Sole proprietorship | 50–249 | Silesian | 20 years | Entrepreneur-manager |
| 3 | Limited liability company | 250 ≤ | Silesian | 17 years | Manager |
| 4 | Public limited company | 250 ≤ | Mazovian | 21 years | Manager |
| 5 | Civil law partnership | 0–49 | Silesian | 19 years | Entrepreneur-manager |
| 6 | Limited liability company | 50–249 | Silesian | 24 years | Manager |
| 7 | Limited liability company | 250 ≤ | Lesser Poland | 19 years | Manager |
| 8 | Public limited company | 250 ≤ | Lesser Poland | 23 years | Manager |
| 9 | Limited liability company | 250 ≤ | Lesser Poland | 17 years | Manager |

## Examination's method and outcome

This examination does not involve any medical research involving human subjects, including research on identifiable human material and data. The consent to the research participation was declared remotely and obtained in the electronic form together with the survey output which was recorded. The survey questionnaire was anonymous thus obtained data was analyzed also anonymously.

**Table 7. Product in the sustainable development perspective.**

| $Q_i^*$ | Answers' characteristics | | | | | | | | | | | | | | $MdeR_i$ |
|---|---|---|---|---|---|---|---|---|---|---|---|---|---|---|---|
| $Q_1$ | 2 | 2.7% | 11 | 14.9% | 16 | 21.6% | 1 | 1.4% | 26 | 35.1% | 16 | 21.6% | 2 | 2.7% | 242/444 ≈ 0.545 |
| | 0 | 0 | 1 | 11 | 2 | 32 | 3 | 3 | 4 | 104 | 5 | 80 | 6 | 12 | |
| $Q_2$ | 2 | 2.7% | 2 | 2.7% | 3 | 4.1% | 0 | 0% | 21 | 28.2% | 42 | 56.9% | 4 | 5.4% | 326/444 ≈ 0.734 |
| | 0 | 0 | 1 | 2 | 2 | 6 | 3 | 0 | 4 | 84 | 5 | 210 | 6 | 24 | |
| $Q_3$ | 2 | 2.7% | 12 | 16.2% | 9 | 12.2% | 1 | 1.4% | 19 | 25.6% | 30 | 40.5% | 1 | 1.4% | 265/444 ≈ 0.597 |
| | 0 | 0 | 1 | 12 | 2 | 18 | 3 | 3 | 4 | 76 | 5 | 150 | 6 | 6 | |
| $Q_4$ | 19 | 25.6% | 30 | 40.5% | 17 | 23% | 4 | 5.4% | 1 | 1.4% | 2 | 2.7% | 1 | 1.4% | 96/444 ≈ 0.216 |
| | 0 | 0 | 1 | 30 | 2 | 34 | 3 | 12 | 4 | 4 | 5 | 10 | 6 | 6 | |
| $Q_5$ | 5 | 6,8% | 33 | 44.5% | 17 | 23% | 2 | 2.7% | 10 | 13.5% | 5 | 6.8% | 2 | 2.7% | 150/444 ≈ 0.338 |
| | 0 | 0 | 1 | 33 | 2 | 34 | 3 | 6 | 4 | 40 | 5 | 25 | 6 | 12 | |
| $Q_6$ | 11 | 14.9% | 27 | 36.4% | 18 | 24.3% | 3 | 4.1% | 10 | 13.5% | 3 | 4.1% | 2 | 2.7% | 139/444 ≈ 0.313 |
| | 0 | 0 | 1 | 27 | 2 | 36 | 3 | 9 | 4 | 40 | 5 | 15 | 6 | 12 | |
| $Q_7$ | 21 | 28.2% | 37 | 50% | 9 | 12.2% | 2 | 2.7% | 3 | 4.1% | 1 | 1.4% | 1 | 1.4% | 84/444 ≈ 0.189 |
| | 0 | 0 | 1 | 37 | 2 | 18 | 3 | 6 | 4 | 12 | 5 | 5 | 6 | 6 | |
| $Q_8$ | 2 | 2.7% | 5 | 6.8% | 7 | 9.5% | 1 | 1.4% | 18 | 24.3% | 35 | 47.2% | 6 | 8.1% | 305/444 ≈ 0.687 |
| | 0 | 0 | 1 | 5 | 2 | 14 | 3 | 3 | 4 | 72 | 5 | 175 | 6 | 36 | |
| $Q_9$ | 2 | 2.7% | 11 | 14.9% | 16 | 21.6% | 1 | 1.4% | 19 | 25.6% | 20 | 27% | 5 | 6.8% | 252/444 ≈ 0.568 |
| | 0 | 0 | 1 | 11 | 2 | 32 | 3 | 3 | 4 | 76 | 5 | 100 | 6 | 30 | |
| $Q_{10}$ | 1 | 1.4% | 2 | 2.7% | 0 | 0% | 2 | 2.7% | 31 | 41.9% | 35 | 47.2% | 3 | 4.1% | 325/444 ≈ 0.732 |
| | 0 | 0 | 1 | 2 | 2 | 0 | 3 | 6 | 4 | 124 | 5 | 175 | 6 | 18 | |
| $Q_{11}$ | 1 | 1.4% | 0 | 0% | 0 | 0% | 0 | 0% | 10 | 13.5% | 45 | 60.8% | 18 | 24.3% | 373/444 ≈ 0.84 |
| | 0 | 0 | 1 | 0 | 2 | 0 | 3 | 0 | 4 | 40 | 5 | 225 | 6 | 108 | |
| $Q_{12}$ | 4 | 5.4% | 3 | 4.1% | 7 | 9.5% | 0 | 0% | 43 | 58.1% | 13 | 17.5% | 4 | 5.4% | 278/444 ≈ 0.626 |
| | 0 | 0 | 1 | 3 | 2 | 14 | 3 | 0 | 4 | 172 | 5 | 65 | 6 | 24 | |

*Question's numbers relate to question numbers in S1 Table which also contains their essence.

**Table 8. Distribution in the sustainable development perspective.**

| $Q_i^*$ | Answers' characteristics | | | | | | | | | | | | | | MdeR$_i$ |
|---|---|---|---|---|---|---|---|---|---|---|---|---|---|---|---|
| Q19 | 13 | 17.6% | 39 | 52.6% | 19 | 25.7% | 2 | 2.7% | 1 | 1.4% | 0 | 0% | 0 | 0% | 87/444 |
|  | 0 | 0 | 1 | 39 | 2 | 38 | 3 | 6 | 4 | 4 | 5 | 0 | 6 | 0 | ≈ 0.196 |
| Q20 | 3 | 4.1% | 22 | 29.7% | 18 | 24.3% | 2 | 2.7% | 18 | 24.3% | 10 | 13.5% | 1 | 1.4% | 192/444 |
|  | 0 | 0 | 1 | 22 | 2 | 36 | 3 | 6 | 4 | 72 | 5 | 50 | 6 | 6 | ≈ 0.432 |
| Q21 | 3 | 4.1% | 24 | 32.3% | 13 | 17.5% | 3 | 4.1% | 23 | 31.1% | 7 | 9.5% | 1 | 1.4% | 192/444 |
|  | 0 | 0 | 1 | 24 | 2 | 26 | 3 | 9 | 4 | 92 | 5 | 35 | 6 | 6 | ≈ 0.432 |
| Q22 | 0 | 0% | 0 | 0% | 0 | 0% | 2 | 2.7% | 15 | 20.3% | 45 | 60.8% | 12 | 16.2% | 363/444 |
|  | 0 | 0 | 1 | 0 | 2 | 0 | 3 | 6 | 4 | 60 | 5 | 225 | 6 | 72 | ≈ 0.818 |
| Q23 | 0 | 0% | 0 | 0% | 2 | 2.7% | 0 | 0% | 9 | 12.2% | 49 | 66.2% | 14 | 18.9% | 369/444 |
|  | 0 | 0 | 1 | 0 | 2 | 4 | 3 | 0 | 4 | 36 | 5 | 245 | 6 | 84 | ≈ 0.831 |
| Q24 | 0 | 0% | 1 | 1.4% | 3 | 4.1% | 0 | 0% | 24 | 32.3% | 41 | 55.4% | 5 | 6.8% | 338/444 |
|  | 0 | 0 | 1 | 1 | 2 | 6 | 3 | 0 | 4 | 96 | 5 | 205 | 6 | 30 | ≈ 0.761 |
| Q25 | 0 | % | 4 | 5.4% | 12 | 16.2% | 0 | 0% | 35 | 47.3% | 19 | 25.7% | 4 | 5.4% | 287/444 |
|  | 0 | 0 | 1 | 4 | 2 | 24 | 3 | 0 | 4 | 140 | 5 | 95 | 6 | 24 | ≈ 0.646 |
| Q26 | 1 | 1.4% | 1 | 1.4% | 4 | 5.4% | 4 | 5.4% | 28 | 37.8% | 29 | 39.1% | 7 | 9.5% | 320/444 |
|  | 0 | 0 | 1 | 1 | 2 | 8 | 3 | 12 | 4 | 112 | 5 | 145 | 6 | 42 | ≈ 0.721 |
| Q27 | 1 | 1.4% | 24 | 32.3% | 11 | 14.9% | 1 | 1.4% | 22 | 29.7% | 12 | 16.2% | 3 | 4.1% | 215/444 |
|  | 0 | 0 | 1 | 24 | 2 | 22 | 3 | 3 | 4 | 88 | 5 | 60 | 6 | 18 | ≈ 0.484 |
| Q28 | 40 | 54% | 28 | 37.8% | 3 | 4.1% | 2 | 2.7% | 1 | 1.4% | 0 | 0% | 0 | 0% | 44/444 |
|  | 0 | 0 | 1 | 28 | 2 | 6 | 3 | 6 | 4 | 4 | 5 | 0 | 6 | 0 | ≈ 0.099 |

*Questions' number relate to question numbers in S1 Table which also contains their essence.

Overwhelming scientific evidence indicates that the unaided human mind is simply not capable of simultaneous analysis of many different, competing factors and then synthesize the results for the purpose of making a rational conclusion. There is experimental evidence i.e. psychological [31] including the well-known Miller study [32] which put forth the notion that humans are not capable of dealing accurately with more than about seven (±2) things at a

**Table 9. Price in the sustainable development perspective.**

| $Q_i^*$ | Answers' characteristics | | | | | | | | | | | | | | MdeR$_i$ |
|---|---|---|---|---|---|---|---|---|---|---|---|---|---|---|---|
| Q13 | 0 | 0% | 0 | 0% | 0 | 0% | 0 | 0% | 3 | 4.1% | 31 | 41.9% | 4 | 5.4% | 191/444 |
|  | 0 | 0 | 1 | 0 | 2 | 0 | 3 | 0 | 4 | 12 | 5 | 155 | 6 | 24 | ≈ 0.43 |
| Q14 | 0 | 0% | 2 | 2.7% | 1 | 1.4% | 0 | 0% | 9 | 12.2% | 51 | 68.8% | 11 | 14.9% | 361/444 |
|  | 0 | 0 | 1 | 2 | 2 | 2 | 3 | 0 | 4 | 36 | 5 | 255 | 6 | 66 | ≈ 0.813 |
| Q15 | 2 | 2.7% | 4 | 5.4% | 10 | 13.5% | 1 | 1.4% | 36 | 48.6% | 19 | 25.7% | 2 | 2.7% | 278/444 |
|  | 0 | 0 | 1 | 4 | 2 | 20 | 3 | 3 | 4 | 144 | 5 | 95 | 6 | 12 | ≈ 0.626 |
| Q16 | 15 | 20.3% | 44 | 59.3% | 7 | 9.5% | 0 | 0% | 5 | 6.8% | 2 | 2.7% | 1 | 1.4% | 94/444 |
|  | 0 | 0 | 1 | 44 | 2 | 14 | 3 | 0 | 4 | 20 | 5 | 10 | 6 | 6 | ≈ 0.212 |
| Q17 | 16 | 21.6% | 48 | 64.8% | 4 | 5.4% | 0 | 0% | 3 | 4.1% | 2 | 2.7% | 1 | 1.4% | 84/444 |
|  | 0 | 0 | 1 | 48 | 2 | 8 | 3 | 0 | 4 | 12 | 5 | 10 | 6 | 6 | ≈ 0.189 |
| Q18 | 57 | 77% | 13 | 17.6% | 2 | 2.7% | 2 | 2.7% | 0 | 0% | 0 | 0% | 0 | 0% | 23/444 |
|  | 0 | 0 | 1 | 13 | 2 | 4 | 3 | 6 | 4 | 0 | 5 | 0 | 6 | 0 | ≈ 0.052 |

*Question's numbers relate to question numbers in S1 Table which also contains their essence.

**Table 10. Promotion mix in the sustainable development perspective.**

| $Q_i^*$ | Answers' characteristics | | | | | | | | | | | | | | MdeR$_i$ |
|---|---|---|---|---|---|---|---|---|---|---|---|---|---|---|---|
| Q$_{29}$ | 0 | 0% | 6 | 8.1% | 4 | 5.4% | 6 | 8.1% | 39 | 52.7% | 16 | 21.6% | 3 | 4.1% | 286/444 |
| | 0 | 0 | 1 | 6 | 2 | 8 | 3 | 18 | 4 | 156 | 5 | 80 | 6 | 18 | ≈ 0.644 |
| Q$_{30}$ | 9 | 12.2% | 43 | 57.9% | 13 | 17.6% | 5 | 6.8% | 2 | 2.7% | 1 | 1.4% | 1 | 1.4% | 103/444 |
| | 0 | 0 | 1 | 43 | 2 | 26 | 3 | 15 | 4 | 8 | 5 | 5 | 6 | 6 | ≈ 0.232 |
| Q$_{31}$ | 16 | 21.6% | 41 | 55.3% | 8 | 10.8% | 3 | 4.1% | 4 | 5.4% | 1 | 1.4% | 1 | 1.4% | 93/444 |
| | 0 | 0 | 1 | 41 | 2 | 16 | 3 | 9 | 4 | 16 | 5 | 5 | 6 | 6 | ≈ 0.209 |
| Q$_{32}$ | 24 | 32.3% | 39 | 52.7% | 4 | 5.4% | 3 | 4.1% | 1 | 1.4% | 2 | 2.7% | 1 | 1.4% | 76/444 |
| | 0 | 0 | 1 | 39 | 2 | 8 | 3 | 9 | 4 | 4 | 5 | 10 | 6 | 6 | ≈ 0.171 |
| Q$_{33}$ | 22 | 29.7% | 27 | 36.5% | 14 | 18.8% | 3 | 4.1% | 6 | 8.1% | 1 | 1.4% | 1 | 1.4% | 99/444 |
| | 0 | 0 | 1 | 27 | 2 | 28 | 3 | 9 | 4 | 24 | 5 | 5 | 6 | 6 | ≈ 0.223 |

*Questions' numbers relate to question numbers in S1 Table which also contains their essence.

time. Taking into consideration that there are 33 areas of examination in this research, it was decided to apply a technique that empower humans in such situations.

Various examination methods involve either examining and studying some phenomenon from the perspective of its various properties, or studying some phenomenon in relation to other similar phenomena and relating them by making comparisons, and then synthesizing findings and drawing conclusions. The latter method leads directly to the method that involves judgments (relative or absolute) regarding a phenomenon. Because humans can make much better relative than absolute judgments [47–49] a pairwise comparisons method was proposed in order to facilitate the process of relative judgments.

The pairwise comparisons method dates back to the beginning of the 20th century and was firstly applied by Thurstone [47] and its first scientific applications can be found in Fechner [50]. However, the method itself is much older and its idea goes back to Ramon Lull who lived in the end of the 13th century. It is claimed [51] that its popularity comes from an influential paper of Marquis de Condorcet [52] who used this method in the election process where voters rank candidates based on their preference. It has been developed and perfected since then in many papers e.g. [53], however its very recent development [18] deserves special attention, thus, to the best knowledge of the authors, it finds its first time ever real problem application.

The objective of the approach was firstly, to evaluate the importance of each survey's question in order to provide a weighted average *MdeR* for each element of a marketing mix i.e. product–*waMdeR*$^{(pt)}$, price–*waMdeR*$^{(pc)}$, distribution (place)–*waMdeR*$^{(pl)}$, and promotion mix–*waMdeR*$^{(pr)}$; and secondly, under the assumption that all elements of a marketing mix are equally important in the process of sustainable development, to compute and evaluate a total weighted average *MdeR* for performance of all elements of a marketing mix i.e. *waMdeR(t)*.

The importance of particular elements was established with the application of the *QSP Multi Criteria Decision Support Tool* [17] aiding a seminal methodology for pairwise comparisons i.e. a non-heuristic approach to pairwise comparisons technique with verifiable accuracy and reliability [18] which for brevity will not be discussed here. Firstly, importance of all the questions was examined from the perspective of their impact on performance of the particular element of marketing mix. Due to a high number of elements, it was done indirectly through intensities which importance was firstly examined in the *QSP Multi Criteria Decision Support Tool* [17].

**Table 11. Alternative's weights and mean errors.**

| Rank | Alternative (intensity) | Weight ± *absolute [relative]* error = Actual weight |
|------|-------------------------|------------------------------------------------------|
| 1 | Extreme | 0.283 ± 0.016[± 5.65%] = [0.267–0.298] |
| 2 | High | 0.236 ± 0.009[± 3.81%] = [0.227–0.244] |
| 3 | Moderate | 0.191 ± 0.010[± 5.24%] = [0.180–0.201] |
| 4 | Slight | 0.161 ± 0.006[± 3.73%] = [0.155–0.167] |
| 5 | Tiny | 0.129 ± 0.008[± 6.20%] = [0.121–0.137] |

The group of experts, whose characteristics are shown in Table 6, evaluated, in pairwise mode, the importance of the following intensities: extreme, high, moderate, slight, and tiny. The questions which the group of experts analyzed were stated as follows: "Taking into consideration the sustainable marketing mix perspective, how much more its particular element's performance evaluation would be affected by the '*e.g. extremely*' significant question in comparison to the '*e.g. moderately*' significant question". Preferences of the group towards those intensities were established on the basis of the experts individual judgments which were geometrically averaged [54] and processed in the *QSP Multi Criteria Decision Support Tool* [17]. It was decided that, among other available scales [55], the flatten numerical preference scale would be applied during the judgment process which comprises of numbers—and their reciprocals—from one (equivalent to the verbal judgment—'equally preferred') to two (equivalent to the verbal judgment—'extremely preferred')–Eq 4:

$$\Im(s) = \begin{cases} f(s) + 1 & s \geq 0, \\ \dfrac{1}{1 - f(s)} & s < 0, \end{cases} \tag{4}$$

where: $f(s) = 0.125s$ for $s \in \{-8, \ldots, 8\} \wedge s \in I$.

The above proposed scale was found to be more appropriate from the perspective of possible estimation errors of intensities weights [56]. The outcome of the entire examination is presented below—Tables 11–13 and Fig 1.

Due to evaluation of intensities relations, evaluation of each survey's question importance became possible. Thus, the importance of each question was evaluated from the perspective of its impact on performance of the particular element of marketing mix. The directive which the group of experts obtained in this stage of evaluation was stated as follows: "Please provide the

**Table 12. Ranking probabilities of compared alternatives.**

| Compared alternatives | Extreme > High | High > Moderate | Moderate > Slight | Slight >Tiny |
|-----------------------|----------------|-----------------|-------------------|--------------|
| Ranking probability | 100.0% | 100.0% | 100.0% | 100.0% |

**Table 13. Relative reciprocal weights for compared intensities.**

| | Extreme | High | Moderate | Slight | Tiny |
|---------|---------|-------|----------|--------|-------|
| Extreme | **1.000** | 1.250 | 1.560 | 1.750 | 2.0 |
| High | 0.800 | **1.000** | 1.310 | 1.450 | 1.810 |
| Moderate | 0.641 | 0.763 | **1.000** | 1.250 | 1.560 |
| Slight | 0.571 | 0.690 | 0.800 | **1.000** | 1.310 |
| Tiny | 0.500 | 0.552 | 0.641 | 0.763 | **1.000** |

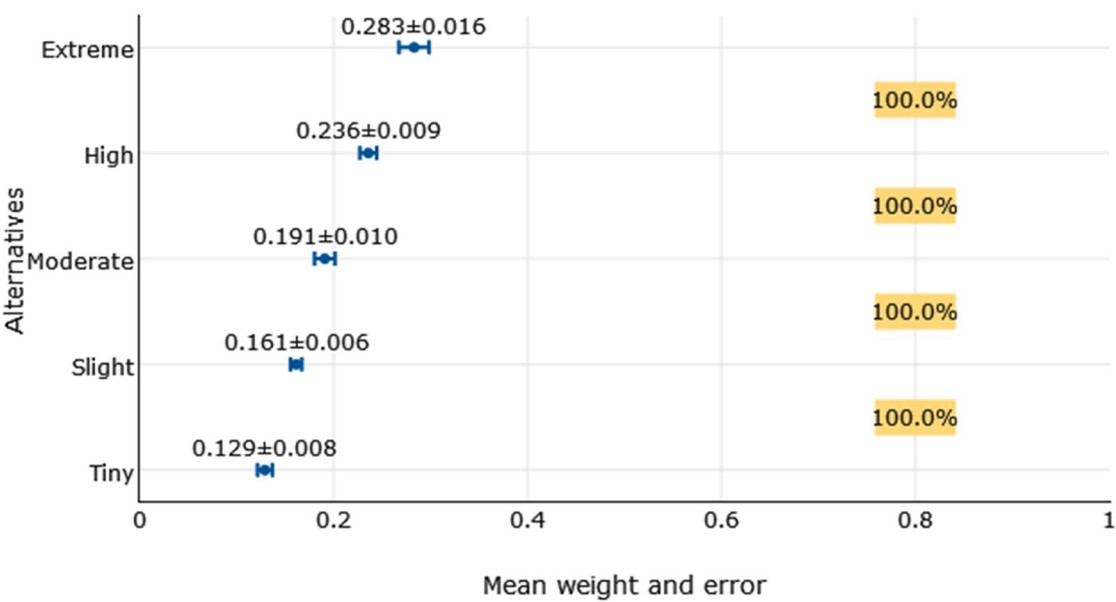

**Fig 1. Intensities (alternatives) comparison results with their mean weight and absolute error.**

significance for the given question from the perspective of its impact on the particular element of the sustainable marketing mix performance evaluation. Please apply for this purpose the following intensities: extreme, high, moderate, slight, and tiny. In this way calculation of $waMdeR^{(pt)}$, $waMdeR^{(pc)}$, $waMdeR^{(pl)}$, and $waMdeR^{(pr)}$ also became possible. The following Tables 14–17 present the outcome of this examination.

**Table 14. Input for $waMdeR^{(pt)}$ calculation.**

| $Q_i^*$ | MdeR$_i$ | Importance (*weight*) | | | waMdeR$_i$[2]x[4] |
|---|---|---|---|---|---|
| | | Verbal | numerical [w$_i$] | $^{(\#)}$rE$_i$[±] | |
| [1] | [2] | [3] | [4] | [5] | [6] |
| Q$_1$ | 0.545 | extreme | 0.283 | 5.65% | 0.154235 |
| Q$_2$ | 0.734 | high | 0.236 | 3.81% | 0.173224 |
| Q$_3$ | 0.597 | high | 0.236 | 3.81% | 0.140892 |
| Q$_4$ | 0.216 | moderate | 0.191 | 5.24% | 0.041256 |
| Q$_5$ | 0.338 | high | 0.236 | 3.81% | 0.079768 |
| Q$_6$ | 0.313 | high | 0.236 | 3.81% | 0.073868 |
| Q$_7$ | 0.189 | moderate | 0.191 | 5.24% | 0.036099 |
| Q$_8$ | 0.687 | extreme | 0.283 | 5.65% | 0.194421 |
| Q$_9$ | 0.568 | extreme | 0.283 | 5.65% | 0.160744 |
| Q$_{10}$ | 0.732 | moderate | 0.191 | 5.24% | 0.139812 |
| Q$_{11}$ | 0.840 | moderate | 0.191 | 5.24% | 0.160440 |
| Q$_{12}$ | 0.626 | slight | 0.161 | 3.73% | 0.119566 |
| SUM: | | | 2.718 | 56.88% | 1.474325 |
| waMdeR$^{(pt)}$ = 1.474325/2.718 = 0.5424 | | | | $^{(\$)}$MrE = 4.74% | |

**Table 15. Input for *waMdeR*$^{(pc)}$ calculation.**

| $Q_i^*$ | MdeR$_i$ | Importance (*weight*) | | $^{(\#)}$rE$_i$[±] | waMdeR$_i$[2]x[4] |
| --- | --- | --- | --- | --- | --- |
| | | verbal | numerical [w$_i$] | | |
| [1] | [2] | [3] | [4] | [5] | [6] |
| Q$_{13}$ | 0.430 | moderate | 0.191 | 5.24% | 0.082130 |
| Q$_{14}$ | 0.813 | moderate | 0.191 | 5.24% | 0.155283 |
| Q$_{15}$ | 0.626 | high | 0.236 | 3.81% | 0.147736 |
| Q$_{16}$ | 0.212 | extreme | 0.283 | 5.65% | 0.059996 |
| Q$_{17}$ | 0.189 | extreme | 0.283 | 5.65% | 0.053487 |
| Q$_{18}$ | 0.052 | extreme | 0.283 | 5.65% | 0.014716 |
| SUM: | | | 1.467 | 31.24% | 0.513348 |
| waMdeR$^{(pc)}$ = 0.513348/1.467 = 0.3499 | | | | $^{(\$)}$MrE = 5.207% | |

**Table 16. Input for *waMdeR*$^{(pl)}$ calculation.**

| $Q_i^*$ | MdeR$_i$ | Importance (*weight*) | | $^{(\#)}$rE$_i$[±] | waMdeR$_i$[2]x[4] |
| --- | --- | --- | --- | --- | --- |
| | | verbal | numerical [w$_i$] | | |
| [1] | [2] | [3] | [4] | [5] | [6] |
| Q$_{19}$ | 0.196 | extreme | 0.283 | 5.65% | 0.055468 |
| Q$_{20}$ | 0.432 | extreme | 0.283 | 5.65% | 0.122256 |
| Q$_{21}$ | 0.432 | extreme | 0.283 | 5.65% | 0.122256 |
| Q$_{22}$ | 0.818 | high | 0.236 | 3.81% | 0.193048 |
| Q$_{23}$ | 0.831 | high | 0.236 | 3.81% | 0.196116 |
| Q$_{24}$ | 0.761 | moderate | 0.191 | 5.24% | 0.145351 |
| Q$_{25}$ | 0.646 | moderate | 0.191 | 5.24% | 0.123386 |
| Q$_{26}$ | 0.721 | slight | 0.161 | 3.73% | 0.116081 |
| Q$_{27}$ | 0.484 | slight | 0.161 | 3.73% | 0.077924 |
| Q$_{28}$ | 0.099 | high | 0.236 | 3.81% | 0.023364 |
| SUM: | | | 2.261 | 46.32% | 1.17525 |
| waMdeR$^{(pl)}$ = 1.17535/2.261 = 0.5198 | | | | $^{(\$)}$MrE = 4.632% | |

**Table 17. Input for *waMdeR*$^{(pr)}$ calculation.**

| $Q_i^*$ | MdeR$_i$ | Importance (*weight*) | | $^{(\#)}$rE$_i$[±] | waMdeR$_i$[2]x[4] |
| --- | --- | --- | --- | --- | --- |
| | | verbal | numerical [w$_i$] | | |
| [1] | [2] | [3] | [4] | [5] | [6] |
| Q$_{29}$ | 0.644 | high | 0.236 | 3.81% | 0.151984 |
| Q$_{30}$ | 0.232 | high | 0.236 | 3.81% | 0.054752 |
| Q$_{31}$ | 0.209 | extreme | 0.283 | 5.65% | 0.059147 |
| Q$_{32}$ | 0.171 | high | 0.236 | 3.81% | 0.040356 |
| Q$_{33}$ | 0.223 | extreme | 0.283 | 5.65% | 0.063109 |
| SUM: | | | 1.274 | 22.73% | 0.369348 |
| waMdeR$^{(pr)}$ = 0.369348/1.274 = 0.2899 | | | | $^{(\$)}$MrE = 4.546% | |

where:

$^{(\#)}$rE$_i$ denotes a relative error of $i^{th}$ question's weight [$w_i$];

$^{(\$)}$ *MrE* denotes the arithmetic mean of relative errors of all weights; and Q$_i^*$ denotes a question's number which relates to question numbers in S1 Table thatalso contains their essence.

Product is the most important element of marketing mix, which is the starting point for defining the other elements of the marketing mix. Manufacturing confectionery products and their consumption affect the natural environment and society. Thus, the questionnaire questions concerned the production area and the impact of this process on the natural environment (seven questions) and the effects of the consumption of confectionery products, which has a significant impact on the health condition of societies (five questions). In order to obtain in-depth information in both, the environmental and social area, the authors found it appropriate to formulate totally twelve questions.

The second element in terms of the importance of impact on sustainable development is distribution. The distribution of confectionery products is, first and foremost, a nuisance to the environment and to a lesser extent to societies. Exhaust and soot emissions, noise, traffic congestion, road traffic jams, pavement damage, road accidents and nuisance for urban and rural residents living near highways are the main problems of unsustainable product distribution. The formulated ten questions allowed the authors of the article to obtain satisfactory information on the distribution of confectionery products.

The number of six questions related to price concerns considering the costs associated with the negative impact of the production and consumption of confectionery products by the confectionery industry. Questions concerning price were intended to provide information as to whether the price reflects all costs—the costs of extracting raw materials, energy, transport and storage, the costs of contributing to global warming and climate change. The six questions asked in the questionnaire allowed the authors to obtain satisfactory information within this area to a exhausting extent.

The last element of the marketing mix is promotion. Promotional messages considered as sustainable should have features that were included by the authors of the article in five questions. Promotional messages do not have a direct negative impact on the natural environment and society. The authors' intention was to examine whether the promotional content provided was in line with the idea of sustainable development and whether it promoted the concept. Such features of promotional messages as transparency, ecological education or information on the activity of enterprises for environmental protection and improvement of the quality of life of societies bear the features of sustainable promotion.

Due to the different number of questions applied to individual elements of the sustainable marketing mix, and thus the importance (weight) of particular marketing mix elements, it was decided to assign weights to individual elements of the marketing mix proportionally to the number of questions. Under this assumption i.e. that all elements of marketing mix have some designated importance in the process of sustainable development, computation of *waMdeR(t)* is presented in Table 18.

Noticeably the total weighted average *MdeR* for performance of all marketing mix elements i.e. *waMdeR(t)* cannot be considered as satisfactory from the perspective of sustainable

**Table 18. Input for *waMdeR(t)* calculation.**

| Element of marketing mix | Importance (*weight*) | waMdeR[(i)] | MrE[(i)] |
|---|---|---|---|
| Product | 12/33 | 0.5424 | 4.740% |
| Price | 6/33 | 0.3499 | 5.207% |
| Distribution | 10/33 | 0.5198 | 4.632% |
| Promotion mix | 5/33 | 0.2899 | 4.546% |
| waMdeR(t) = 0.4623 | | *MrE(t) = 4.78125% | |

*****MrE(t)** denotes the arithmetic mean of mean relative errors for *waMdeR[(i)]*.

development of the confectionery industry, especially for a developed country like Poland. It should be underlined that this ratio is burdened with estimation error. However, taking into account that the assumed level of accuracy $d \leq 0.08$ for the survey assumes less than plus minus eight percent of relative error margin, and the arithmetic mean of mean relative errors for $waMdeR^{(i)}$ i.e. $MrE(t) = 4.78\%$, the total mean relative estimation error for this examination ($TMeE$) should not exceed 13.164%, accurately $TMeE<0.1316375$. Thus, the following estimation interval for this examination can be presented as $0.401444 \leq waMdeR(t) \leq 0.523156$. Then, the best situation $waMdeR(t) \approx 0.5232$, still presents a rather poor sustainable marketing mix performance in relation to sustainable development of the Poland's confectionery industry. Noticeably, the ultimate value of $waMdeR(t)$ depends significantly on the assumed importance of considered elements of the sustainable marketing mix.

As the examination fundamentally is based on qualitative data obtained from survey's questionnaire which quantifies it through the elaborated methodology it was decided to support the above presented research outcome with a compact case-based quantitative analysis of sustainable performance for a few selected international companies also operating on Poland's market.

Thus, the percentage values of the achieved effects from the implementation of sustainable activities in the area of confectionery production are presented below in Table 19. Those values concern the period of the last two years i.e. 2018–2019, and come from official reports published on the websites of companies [37–39,57–60].

The first analyzed factor is the reduction of CO2 emissions in the production activities of enterprises. The emissions reduction ranges from 3.5% for Mars to 40% for Ferrero. The presented values are characterized by a large range. In other enterprises, the reduction of CO2 emissions in production processes ranges from 14–15% to 25–34%. In this case, the presented values are at a similar level. Note the performance of Nestle—34% and Ferrero at 40%, which are significant figures compared to the rest of the companies.

Another important factor taken into account in the sustainable production activities of selected companies is the use of water in production processes. The implemented pro-environmental measures allow to reduce water consumption from 10% in companies such as Mondelez and Lindt, up to 20% in Mars and 35% in Nestle. The indicated values are at a similar level and do not differ significantly from each other.

In the confectionery production processes, in addition to CO2 emissions and water consumption, occurs also production of manufacturing waste. The data for selected business units indicates that the level of reduction of production waste ranges from 20% to 96%. There is a quite big discrepancy here. Mondelez declares a waste reduction of 20%, Lindt 50%, Cemoi Chocolatier 90%, and Nestle 96% waste reduction, which should be emphasized. This is a very

**Table 19. Effects in percentage points of the implementation of sustainable activities in selected companies of the confectionery industry for 2018–2019.**

| Firm's Name | Mondelez | Ferrero | Mars | Lindt | Cemoi Chocolatier | Nestle | Maspex |
|---|---|---|---|---|---|---|---|
| Reduction of CO2 emissions from production activities | 15 | 40 | 3.5 | 14 | – | 34 | 25 |
| Water consumption reduction | 10 | – | 20 | 10 | – | 35 | – |
| Reducing the amount of waste in production process | 20 | – | – | 50 | 90 | 96 | – |
| Own production of electricity from renewable sources | – | 18 | – | 39 | 30 | – | 15 |
| The use of packaging made fully or partially of recycled raw materials | 90 | 100 | 90 | 90 | 78 | 87 | 97 |
| Sugar reduction | 1 | – | – | – | 20 | 5 | – |
| Palm oil reduction | 1 | – | – | – | 90 | 15 | – |

Source: [37–39,57–60].

good result that allows us to conclude that Nestle strives to completely eliminate production waste accompanying the confectionery production processes.

The fourth factor that takes into account sustainable production activities in the confectionery industry is obtaining electricity from renewable sources. The achieved values range from 15–18% (Maspex/Ferrero) to 30–39% (Cemoi Chocolatier/Lindt). None of the analyzed enterprises exceeded the level of 50% of obtaining electricity from renewable sources used in production processes. This means that companies in the confectionery industry mainly use energy obtained from fossil fuels, which leaves the so-called carbon footprint.

A serious problem in the production activities of the confectionery industry and, more broadly, the food industry is the issue of packaging. For this reason, enterprises undertake activities aimed at minimizing the negative impact of packaging on the natural environment. Such an activityis the use of packaging made of fully or partially recycled raw materials. The obtained data jump to a high level of these activities, ranging from 78% to 100% of the used packaging from recycled raw materials. The use of packaging at the level of 100% was declared by Ferrero, the result was 97% by Maspex, 90% by Mondelez, Mars, Lindt, 87% by Nestle and 78% by Cemoi Chocolatier.

In addition to environmental issues, the sustainable production activities of the confectionery industry also include social issues, including the impact of confectionery consumption on health. The main ingredient used in the production of confectionery products is sugar. The available data shows that the reduction of sugar in products ranges from 5% for Nestle to 20% for Cemoi Chocolatier.

Another negative ingredient used in the production of confectionery products is palm oil. The reduction of palm oil at Cemoi Chocolatier is as much as 90%, while Nestle declares only 15% unit reduction of palm oil in its confectionery products, while at Mondelez, the reduction of sugar and oil is merely1%.

## Discussion

It is claimed by various scientists and researchers [61–66] that there has been a growing interest in corporate social responsibility (CSR) and sustainable marketing, including sustainable marketing mixin the last decade. However, the literature research carried out by the authors of this article indicates a scarcity of research concerning sustainable marketing. Only a few books [2,12,67–72] and a few relatively recent research papers [16,73–77] devoted to this concept can be recognized herein. The modest scientific achievements in the field of sustainable marketing and sustainable marketing mix are not the result of downplaying or disregarding this area of knowledge. The reasons for this situation should be seen in the fact that sustainable marketing, which includes sustainable marketing mix, is a new and emerging area of science that requires further supplementation and improvement through both theoretical and empirical research. The poor state of knowledge indicates the emergence of research problems that require identification and analysis. When reviewing the literature in terms of sustainable marketing mix of the food and drinkindustry, one should recognize the impact of Rudawska's publication [78]. The study presents an analysis of the concept of sustainable marketing forthe food and drink industry enterprises in six European countries using the extended formula from 4P to 5P including employees.

Sustainable marketing mix of consumption industry enterprises includes four instruments of marketing impact on customers—sustainable product, price, distribution and promotion mix. Appropriate management of marketing mix tools by managers of consumption industry enterprises is aimed at ensuring not only economic benefits, but also environmental and social goals. Taking into account triple bottom line values [9,12,79] by an organization means real

building of financial, natural and social capital based on the concept of sustainable development [80]. However, researchers' observations show that triple bottom line is not a universally dominant business model in manufacturing companies [16,81–83].

The main tool in the strategy of sustainable marketing mix of consumption industry enterprises is the product. The essence of a sustainable product is determined by Fuller [72]. The author claims that the sustainable product has ecological features that were created thanks to decisions on the method of production, use of materials from which the product was made, mode of operation, time of operation, distribution, usage and the possibility of product withdrawal in the final life cycle. The author also describes the sustainable product as *green product*. Martin and Schouten [67] believe that a sustainable product requires responsibility on the part of the company. They say that the term 'sustainable' depends on how the product is managed through its understanding, controlling and communicating of its environmental effects, its lifetime health and safety i.e. from the moment of its manufacturing to discarding or reusing it.

Leitner [70] presents his position on the nature of sustainable products. He believes that a sustainable product is burdening the environment to a lesser extent. Leitner points out that sustainable products that take into account social aspects and customers' needs will better meet the expectations of buyers than conventional offers. Kadirov [71] claims that from the point of view of original system thinking, existing marketing concepts appear to be insufficient. The author points out that many marketing concepts develop alternative frameworks in trading systems. An example of such a system that creates sustainable products is the sale of hybrid cars. Such activities provide an alternative basis for redefining the basic macro marketing problems that should be particularly useful for system designers and decision makers. Belz and Peattie [68,69] also express their view on the essence of a sustainable product. According to them, these are goods that meet the needs of customers and significantly improve social and environmental performance throughout the entire product life cycle compared to conventional or competitive offers. Peattie [83] presents a similar interpretation of a sustainable product.

Sustainable products provide customers satisfaction with their purchase and consumption. Satisfying needs by purchasing a specific consumption product is not just about taste. The buyer who is not indifferent to social aspects, including the impact of consumption on human health, also pays attention to the product's raw material composition, in particular to ingredients and substances affecting his health [1].

The dual concentration of the product covering environmental and social aspects is also important. Food production is accompanied by environmentally burdensome manufacturing processes, distribution of finished products to final destinations, packaging, and possible disposal and recycling options. In addition to the ecological factors indicated, the buyer of the product is not indifferent to social problems that could be associated with food production [84]. Sustainable consumption productsarecharacterized by social properties, such as decent working conditions where the product was produced, a policy of purchasing raw materials for production, and/or a remuneration system for employees. Therefore, double concentration covers the environmental and social values of sustainable products. Food industry companies must constantly provide customers with coherent values and benefits, both ecological and social, which they receive by purchasing said products. This is associated with continuous improvement of the product offer so that it is attractive and strengthens customers in the belief that the products they buy are manufactured with a sense of environment responsibility and are not a cause of social problems. Buyers expect ethical behavior, good quality, safe and healthy products, ecological packaging, reliable information and fair advertising, clear complaint procedures, and response to social needs [85,86].

Continuous improvement of the product offer leads to an increase in the company's market competitiveness. In addition to attractive product prices, innovation, good quality or easy availability at the point of sale, the manufacturer's openness to the environmental and social aspects of offered products will undoubtedly be contributing attributes in achieving competitive advantage. Emphasizing the importance of conducting sustainable production and commercial activities will certainly contribute to achieving planned economic goals [85]. Examination in this paper indicates that importance of conducting sustainable production from the perspective of the Poland's confectionery industry is merely slightly above the middle of the scale from zero to one i.e. 0.5424 with relative error ±13.12%.

An important element in assessing marketing mix ventures in relation to the sustainable development of the consumption industry is the aspect of food prices. Sustainable product pricing is not an easy task, because the price of the product should not only include the manufacturer's revenues and financial outlays incurred in the production of said goods, but also should take into account the environmental and social costs associated with production, sale and disposal of expired or damaged food products. In addition, the difficulty in setting sustainable prices is compounded by the belief that sustainable products must cost more because their production and sales involve environmental and social aspects, which entails additional investments [87]. Consumer objections to higher prices for sustainable products are not fully justified. Buyers of food products objecting to the inclusion of environmental and social costs in the price of the product show an egoistic attitude, considering only their own benefits- in this case financial—forgetting about future generations and their opportunities for development. Martin and Schouten [67] argue that balanced price takes full account of the costs of production and marketing, not only in economic terms, but also in environmental and social terms, ensuring consumer benefit and profit for the producer.

For many consumers of food products, sustainability is a secondary factor and is not taken into account when shopping. In most confectionery companies, product pricing strategies are created for the customers and focused on providing them with financial benefits. Therefore, social and ecological aspects play a small role in the product pricing process. Such actions are particularly visible when the seller is facing the need in maintaining a relatively large market with high and strong competition. The issue of setting sustainable prices for confectionery products raises controversy and is the subject of discussions undertaken by marketing managers and consumers. Usually, higher prices of sustainable confectionery products are also a serious obstacle to the full implementation of sustainable economic activity by confectionery enterprises. Higher prices for sustainable products limit demand which conflicts with the economic interests of food producers. Despite opposition and unfavorable opinions of most consumer groups, efforts should be made to increase interest and accept prices for sustainable products [76]. However, examination in this paper indicates that Poland, as a developed country, cannot be perceived as a good example of marketing mix performance from the perspective of price as one of its elements in the sustainable development of the food industry as represented by Poland's confectionery industry. The examination indicates that this area positions itself below the middle of a scale from zero to one, at a level of 0.3499 with relative error of ±13.63%.

Distribution, as another element of marketing mix, plays a very important role in sustainable marketing activity of consumption industry enterprises. The main task of sustainable distribution of food products is to effectively deliver products desired by buyers to agreed places at the right time, while maintaining product integrity in terms of quality. The essence of sustainable distribution is therefore ecological and social aspects relating to infrastructure, with particular emphasis on activities such as transport and storage [88]. Consumption industry companies have a wide range of instruments at their disposal to implement the concept of

sustainable food distribution. One of the main activities in this area is the selection of sustainable means of transport. Hybrid, electric or hydrogen powered vehicles are considered environmentally friendly and do not cause serious damage to nature. However, the use of these means of transport is not a commonly used practice among food producers in the country under study. The main means of transport are both passenger and heavy motor vehicles powered by liquid fuels such as gasoline and diesel.

Distribution activities focused on sustainable development, regardless of the means of transport used, should assume that the selection of routes will shorten delivery time, which translates into reduction of fuel consumption, thus reducingdelivery cost of consumption products and prices for product purchases in a shop. One way of reducing the negative impact of distribution on the environment is to shorten the distance that the product travels by creating local supply chains. Before a manufactured food product reaches its final recipients, it passes through a complex network of interconnected intermediary links. These cells are grouped in various combinations that form distribution channels [89].

Sustainable distribution of consumption products requires a change in the approach to environmental protection and participants' social attitudesinvolved in distribution processes. Already at the management level, there is a need to change the way of thinking about the essence of distribution in a sustainable aspect. Sustainable distribution, in addition to efficiently delivering products desired by buyers to agreed places at the right time while maintaining product integrity in terms of quality, takes into account ecological and social aspects. One of the ways ofcontributing to sustainable distribution activities is to limit to a necessary minimum the intermediaries involved in delivering the product. Too many intermediaries contribute to the price of the product and expose the product to damage when passing it on to the next intermediary. Means of transport should be modern and interfere with the natural environment as little as possible. Hybrid or electric vehicles do not cause emissions to the atmosphere and are environmentally friendly. The social and ecological sensitivity of people involved in the movement of goods is also important for distribution processes [77,90]. Examination in this paper indicates that importance of conducting sustainable distribution from the perspective of the Poland's confectionery industry is slightly above the middle of the scale from zero to one i.e. 0.5198 with relative error of ±13%.

Sustainable promotion mix is the fourth instrument of sustainable marketing mix which includes advertising, direct marketing, sales promotion, personal promotion and public relations. Sustainable promotion mix should support a company's activities focused on sustainable development. Activating the sale of food products through the use of mix promotion should take into account the triple bottom line values i.e. it should be focused on social, environmental and economic goals. Peattie [83] notes that effective communication is not just a matter of sending positive news about eco-activities. It requires involvement in a dialogue regardinga company's activities and the environment. Fuller [72] associates sustainable marketing promotion mix with marketing communication that serves as information support for various pollution prevention and resource recovery strategies.

The use of sustainable mix promotion by consumption industry companies provides food producers with extensive opportunities to communicate with market participants. Promotional messages addressed to buyers of food products are usually designed to inform and persuade customers to buy, and also fulfill the function of reminding of the manufacturer's offer. The content transmitted in sustainable promotional messages calls for the purchase of pro-ecological and pro-social products, informs the potential buyer about the sustainable nature of the enterprise itself, and also shapes the image of the organization as a friendly and caring company. Promotional messages transmitted through sustainable promotion mix take the

form of appeals. It is claimed [68,69] that every marketing message has an appeal that tries to get recipients to respond. The authors indicate three types of appeals:

- rotational appeal, which may, for example, concern organic food by presenting it as healthier, with low calorie level, making it more economical for the consumer;

- an emotional appeal that seeks to create an emotional bond with the consumer;

- a moral appeal regarding the human sense of right and wrong.

A similar view on advertising appeals is expressed by Martin and Schouten [67] which distinguish rational, emotional and hybrid appeals resulting from a combination of rational and emotional advertising appeals. Activating sales through a sustainable mix promotion can contribute to more sustainable behavior not only of the consumption industry enterprises, but also of the food buyers themselves. Sustainable enterprise communication focused on selling products not only takes into account the financial aspects of an organization, but also considers environmental and social factors first [82,84,85,87,89–91]. Examination in this paper indicates that this element of marketing mix takes the lowest performance rank in the perspective of sustainable development of the Poland's confectionery industry. The research shows that this area must be evaluated as poorly performed, with its intensity on a scale from zero to one, on the level 0.2899 with relative error of ±12.91%.

At best, the total weighted average mean development ratio for performance of all elements of marketing mix i.e. $waMdeR(t) \approx 0.5232$ presents a rather poor marketing mix performance in relation to sustainable development of the Poland's confectionery industry as a representative of the Polish food market. On the other hand, the analysis of the state of knowledge clearly indicates that more and more space is devoted to the issues of sustainable marketing mix. Issues related to the important area of sustainable marketing mix of food products are particularly important due to their impact on the four components of the marketing composition on the natural environment and society.

## Conclusions

As a result of the researchconcerning the actual situation on thefood market, as well as the study of the literature on the subject and the analysis with evaluation of empirical research within Poland's confectionery industry, the objective of the research in this paper has been achieved.

Theoretical considerations, conclusions and analysis of research results contained in the research paper do not fully solve the subject matter. Therefore, the search for new and creative solutions aimed at reconciling the economic development of food industry enterprises as well as other economic entities with ecological and social values remains an open issue [92–94]. Sustainable marketing mix, as one of the elements of sustainable marketing, is in the developmental phase and requires further scientific exploration and work aimed at improving its procedures [15,95].

The article is addressed to practitioners and theoreticians dealing with the issues of sustainable marketing, with particular emphasis on sustainable marketing mix. The paper focuses on both theoretical and practical areas of consumption products marketing mix based on the principles of sustainable development. In accordance with the adopted research objective, the research hypothesis was formulated and evaluated. The empirical research conducted within enterprises of the confectionery industry confirmed the research hypothesis. The study may serve as support for food industry enterprises and manufacturing enterprises operating in other sectors of the economy in pursuit of set economic, environmental and social goals.

## Supporting information

**S1 Table. The Questionnaire's template.**
(PDF)

**S2 Table. Database Q$_1$-Q$_{12}$.** $Q_i$ denotes question's number which relates to question numbers in S1 Table that also contains their essence.
(PDF)

**S3 Table. Database Q$_{13}$-Q$_{18}$& Q$_{29}$-Q$_{33}$.** $Q_i$ denotes question's number which relates to question numbers in S1 Table that also contains their essence.
(PDF)

**S4 Table. Database Q$_{19}$-Q$_{28}$.** $Q_i$ denotes question's number which relates to question numbers in S1 Table that also contains their essence.
(PDF)

## Acknowledgments

The authors would like to thank two anonymous Referees for their benevolence and Editors for their support in meeting previously unprecedented high standards set up for papers submissions.

## Author Contributions

**Conceptualization:** Pawel Tadeusz Kazibudzki, Tomasz Witold Trojanowski.

**Data curation:** Tomasz Witold Trojanowski.

**Formal analysis:** Pawel Tadeusz Kazibudzki, Tomasz Witold Trojanowski.

**Investigation:** Pawel Tadeusz Kazibudzki.

**Methodology:** Pawel Tadeusz Kazibudzki, Tomasz Witold Trojanowski.

**Resources:** Tomasz Witold Trojanowski.

**Writing – original draft:** Pawel Tadeusz Kazibudzki, Tomasz Witold Trojanowski.

**Writing – review & editing:** Pawel Tadeusz Kazibudzki.

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
