## [Decision Letter · Decision Letter 0]

10 Sep 2020

PONE-D-20-18328

Examination of marketing mix performance in relation to sustainable development of the Poland’s confectionery industry.

PLOS ONE

Dear Dr. Kazibudzki,

Thank you for submitting your manuscript to PLOS ONE. After careful consideration, we feel that it has merit but does not fully meet PLOS ONE’s publication criteria as it currently stands. Therefore, we invite you to submit a revised version of the manuscript that addresses the points raised during the review process.

Please revise this paper according to the reviewers' comments and suggestions.

We look forward to receiving your revised manuscript.

Kind regards,

Isabel Novo-Cortí

Academic Editor

PLOS ONE

Journal Requirements:

2. Please provide additional details regarding participant consent. In the ethics statement in the Methods and online submission information, please ensure that you have specified (a) whether consent was informed and (b) what type you obtained (for instance, written or verbal, and if verbal, how it was documented and witnessed). If your study included minors, state whether you obtained consent from parents or guardians. If the need for consent was waived by the ethics committee, please include this information.

Reviewers' comments:

Reviewer's Responses to Questions

**Comments to the Author**

1. Is the manuscript technically sound, and do the data support the conclusions?

Reviewer #1: Yes

Reviewer #2: Yes

2. Has the statistical analysis been performed appropriately and rigorously? 

Reviewer #1: Yes

Reviewer #2: Yes

3. Have the authors made all data underlying the findings in their manuscript fully available?

Reviewer #1: Yes

Reviewer #2: Yes

4. Is the manuscript presented in an intelligible fashion and written in standard English?

Reviewer #1: Yes

Reviewer #2: Yes

5. Review Comments to the Author

Reviewer #1: - Interesting work, which deals with the evaluation of sustainability in the marketing mix policies in the confectionary industry in Poland.

- Beyond the specific sector in which this study is situated, the relevance lies in the novelty of this research in the literature.

- It addresses the essential topic of sustainability in terms of its relationship to product, price, promotion and placement policies, both from a theoretical point of view and through an elaborate empirical study based on the calculation of the Mean Development Ratio for each of the 4 P's.

- Although there are previous studies that address this question, this research, in the opinion of whoever writes this, goes one step further by preparing a powerful questionnaire with 33 items and a Likert scale supplied to 74 companies in the sector and with the methodological development that it carries out.

- The anti-plagiarism program Turnitin has been passed and the work is original.

- The article in English is well written at the grammatical and linguistic level.

- The bibliography is rich in researches on management, sustainability, marketing and methodology.

- It is recommended to add bibliographic citations in the first paragraph of point 2.2. where health claims are made, linked to confectionary without theoretical support.

Reviewer #2: The article is very interesting. The theme is very current. Sustainable marketing is of great interest.

I have accepted that this article be published because of the contribution it can make to the literature on the subject, although the survey can be improved.

I think that the survey would need to include next to each question, when possible, the same question but formulated in quantitative terms, asking the company to provide a quantitative value. These new questions will serve to see if there is congruence with the answer, on a scale from 0 to 6, in the previous question. For example:

Along with question number 2:

2. The company consciously saves energy, water and fuels needed for production.

The quantitative question could be formulated:

2.b Indicate the percentage of savings that the company has had in the last two years in:

Energy ....%

Water...%

Fuel ...%

To the question:

3. The company eliminates or limits the emission of harmful gases, dusts, fragrances, industrial wastewater, and production waste.

The quantitative question would be:

3b. Indicate what average reduction (in%) your company has made in recent years in its production in:

Harmful gas reduction ...%

Dust reduction ...%

Reduction of industrial wastewater ...%

Production waste reduction ...%

These additional, more quantitative questions will serve to homogenize and verify the reliability of the answers, by the companies, in the questions with scales from 0 to 6. In addition, they open new avenues of investigation to advance in the knowledge of how companies carry out or not sustainable marketing.

6. PLOS authors have the option to publish the peer review history of their article (what does this mean?). If published, this will include your full peer review and any attached files.

Reviewer #1: No

Reviewer #2: No

---

## [Author Response · Author response to Decision Letter 0]

2 Oct 2020

Dear Professors – the order of our answers for your reviews was decided to be as in the editorial system of PLOS ONE and the email we received from the Editorial Office. As we are unaware of your identities we will call you Reviewer No. 1 and Reviewer No. 2. The numbers standing next to your anonymous identity are assigned for the order of your appearance in the editorial system of the Journal and our responses identification and have no other meaning whatsoever.

Reviewer No. 1

Dear Reviewer – we would like to thank you for your time and appreciation of our research efforts. We owe you so much for your exceptionally positive judgment of our examination. Obviously, we fully fulfilled your suggestion to add some references in Section 2.2 of the research paper. They are introduced as positions 37-39 and 40-46. Thank you so much for your benevolence! 

Reviewer No. 2

Dear Reviewer – we would like to thank you for your time and appreciation of our research efforts. We truly appreciate your very positive judgment of the paper. Once we saw your suggestion to improve the survey we thought we wished we had known it before we have done our research. At the present moment the survey is done and, for a few reasons but mostly financial, there is no way of its complete iteration. The problem seemed to be a dead end – but, at the end, it presented itself as the opportunity. Now, we know that our research benefited better in the way the suggestion was obtained i.e. after our examination was completed. We will explain it now.

The paramount reason why we applied in our research the methodology which enables quantification of qualitative attributes which cannot be quantified otherwise was the statement attributed to Albert Einstein: “Not everything what counts can be counted, and not everything what can be counted – counts”.

Appreciating your suggestion for numerical support of our research we carefully analyzed all survey questions in the questionnaire for the possibility of adding the additional quantitative questions as you had proposed. We realized then that only 6 of them, out of total 33 in the survey questionnaire could be improved in the way you suggested. Noticeably, these 6 questions constitute a rather small fraction of all 33 survey questions. Anyway, we decided to contact selected companies which formed our primary research sample and find out if they can help us to collect necessary data. We contacted 40 out of 74 previously selected companies operating within the confectionery industry in Poland and asked their representatives about quantifiable data concerning those eight questions which, in our opinion, could be numerically supported. It occurred, that all of those companies, except 2 of them, do not collect data we were asking for. Then, we realized on this bases that 95% of all companies in the sample presumably could not provide any quantitative information we could use, thus its statistical examination in such a case would be pointless. So, we decided to search some statistics for the industry. Unfortunately, we also disappointed ourselves in this matter because there are no statistics for confectionery industry in Poland within the area of our interest. Although we found some statistics concerning the issue, they concerned the entire food industry in Poland what seemed for us ambiguous from the perspective of our examination. Eventually, we realized that the only solution we can provide to meet your suggestion to improve our research numerically is to present some statistics, as a secondary data, concerning sustainable performance of a few selected most recognizable international companies operating as well in the Poland’s confectionery industry. The data together with our comment are introduced at the end of Methodological Section in lines 508-566 of the manuscript with marked changes, and in lines 506-564 of the unmarked revised manuscript.

We can only hope that our analysis built on those examples can be accepted as the case-based research that significantly improves our research numerically and, although partially, meet your suggestion to support our research with some quantitative data. Please, accept our apologies that we are unable to fully implement your proposition of our survey improvement.

Respectfully yours,

Pawel Tadeusz Kazibudzki, PhD

(on behalf of all authors)

---

## [Editor Report · Decision Letter 1]

6 Oct 2020

Examination of marketing mix performance in relation to sustainable development of the Poland’s confectionery industry.

PONE-D-20-18328R1

Dear Dr. Kazibudzki,

We’re pleased to inform you that your manuscript has been judged scientifically suitable for publication and will be formally accepted for publication once it meets all outstanding technical requirements.

Kind regards,

Isabel Novo-Cortí

Academic Editor

PLOS ONE
---

## [Editor Report · Acceptance letter]

14 Oct 2020

PONE-D-20-18328R1 

Examination of marketing mix performance
in relation to sustainable development
of the Poland’s confectionery industry 

Dear Dr. Kazibudzki:

I'm pleased to inform you that your manuscript has been deemed suitable for publication in PLOS ONE. Congratulations! Your manuscript is now with our production department. 

Kind regards, 

on behalf of

Dr. Isabel Novo-Cortí 

Academic Editor

PLOS ONE